# Mutational Profile of Blood and Tumor Tissue and Biomarkers of Response to PD-1 Inhibitors in Patients with Cutaneous Squamous Cell Carcinoma

**DOI:** 10.3390/cancers17071172

**Published:** 2025-03-31

**Authors:** Mark J. Chang, Daniel B. Stamos, Cetin Urtis, Nathan L. Bowers, Lauren M. Schmalz, Logan J. Deyo, Martin F. Porebski, Abdur Rahman Jabir, Paul M. Bunch, Thomas W. Lycan, Laura Buchanan Doerfler, Hafiz S. Patwa, Joshua D. Waltonen, Christopher A. Sullivan, J. Dale Browne, Wei Zhang, Mercedes Porosnicu

**Affiliations:** 1Department of Internal Medicine, Wake Forest University School of Medicine, Winston-Salem, NC 27157, USA; mjchang@wakehealth.edu (M.J.C.); dstamos@wakehealth.edu (D.B.S.); ldeyo@wakehealth.edu (L.J.D.); mporebski@wakehealth.edu (M.F.P.); ajabir@wakehealth.edu (A.R.J.); tlycan@wakehealth.edu (T.W.L.J.); 2Center for Cancer Genomics and Precision Oncology, Wake Forest University School of Medicine, Winston-Salem, NC 27157, USA; curtis@wakehealth.edu (C.U.); lschmalzmd@gmail.com (L.M.S.); wezhang@wakehealth.edu (W.Z.); 3Wake Forest Baptist Comprehensive Cancer Center, Winston-Salem, NC 27157, USAhpatwa@wakehealth.edu (H.S.P.); jwaltone@wakehealth.edu (J.D.W.); csulliva@wakehealth.edu (C.A.S.); jdbrowne@wakehealth.edu (J.D.B.); 4Knoxville Institute of Dermatology, Knoxville, TN 37919, USA; nbowers@dermatologyknoxville.com; 5Department of Radiology, Wake Forest University School of Medicine, Winston-Salem, NC 27157, USA; 6Department of Dermatology, Wake Forest University School of Medicine, Winston-Salem, NC 27157, USA; ldoerfle@wakehealth.edu; 7Department of Otolaryngology, Wake Forest University School of Medicine, Winston-Salem, NC 27157, USA

**Keywords:** cutaneous squamous cell carcinoma (cSCC), biomarkers, checkpoint inhibitor, immunotherapy, PD-1 inhibitor, PD-L1, tumor mutational burden (TMB), genomics, tDNA, ctDNA

## Abstract

PD-1 checkpoint inhibition produces durable responses in cutaneous squamous cell carcinoma, yet there are no validated biomarkers predictive of response. From a genomic standpoint, a couple of studies have demonstrated response association to an inflamed gene signature or copy gain in chromosome 3q. This study analyzes tDNA and ctDNA genomic profiles and identifies eight mutated genes, *CDK12*, *CTCF*, *CTNNB1*, *IGF1R*, *IKBKE*, *MLH1*, *QKI*, and *TIPARP*, as possible predictors for non-responders to PD-1 inhibitors. These findings propose novel biomarkers that have not yet been reported. These results are impactful in the development of personalized oncology and patient selection for monotherapy with PD-1 inhibitors in cSCC patients. Continued work is needed for biomarker validation and for incorporating these findings into clinical practice.

## 1. Introduction

Cutaneous squamous cell carcinoma (cSCC) is the second most common skin cancer, and pathogenesis is highly associated with aging, immunosuppression, and cumulative UV exposure [1,2]. Due to these risk factors, cSCC is associated with increased DNA damage, making it one of the most genetically altered cancers [3]. Up to 5% of patients have locoregionally advanced disease with high risk of recurrence and metastasis [4,5,6]. Furthermore, there is a higher propensity of the cancer in the head and neck region due to chronic UV radiation [7]. When patients are not candidates for curative surgical resection or radiation therapy, PD-1 inhibitors are the recommended systemic treatment. The emergence and efficacy of PD-1 inhibitors have changed the landscape of treatment and provide an exceptional option for those with advanced disease. In 2019, PD-1 inhibitors became FDA-approved for use in cSCC. Initial phase I/II EMPOWER-cSCC trials (NCT02383212, NCT02760498) established cemiplimab as the first PD-1 inhibitor to gain approval in this disease population. Out of 104 patients, 45% achieved an objective response, with duration of response reaching more than 40 months and median overall survival (OS) reaching more than 48 months [8,9,10,11]. Similarly, in the phase II CARSKIN trial (NCT03284424), treatment with pembrolizumab achieved a 40% objective response rate, median progression-free survival (PFS) of 7 months, and 60% one-year OS. The results of this trial subsequently led to pembrolizumab’s FDA-approval as an alternative PD-1 inhibitor [12,13]. Cemiplimab or pembrolizumab are recommended when curative radiation therapy or surgery are not feasible for advanced disease [14].

The current literature is limited regarding biomarkers that may predict PD-1 inhibitor treatment efficacy in cSCC. To date, possible clinical predictors include primary tumor location, serum absolute lymphocyte count, and serum lactate dehydrogenase levels [15,16,17,18]. At the molecular and genomic level, PD-L1 and TMB are more commonly studied across tumor types. For cSCC, PD-L1 expression is not validated as a biomarker for therapeutic decisions. In the initial landmark trials (EMPOWER-cSCC and CARSKIN), PD-L1 expression did not significantly correlate with treatment response, but higher response rates were observed when PD-L1 was ≥1% [19,20]. It is believed that PD-L1 negative patients may still benefit from checkpoint inhibitors; thus, the treatment is recommended irrespective of expression level [14]. Tumor mutational burden (TMB) has only been investigated in retrospective studies and may be positively correlated to treatment response. Hanna et al. analyzed 55 cSCC patients and reported that higher TMB (median: 25 vs. 11 mut/Mb) was associated with positive response to treatment with a PD-1 inhibitor (cemiplimab, pembrolizumab, or nivolumab) [21]. In et al. analyzed 15 cSCC patients treated with a PD-1 inhibitor (cemiplimab, pembrolizumab, or nivolumab) and found higher TMB levels in responders vs. non-responders (median: 60 vs. 9 mut/Mb) [22]. Less studied genomic biomarkers include copy number alterations (CNA), inflamed immune cell gene signatures, and DNA damage repair gene mutations. Kacew et al. analyzed 16 cSCC patients treated with a PD-1 inhibitor (not specified) and found that a higher frequency of CNAs was present in responders vs. non-responders (median: 97 vs. 47.5) [23]. Furthermore, low-copy gains in the q arm of chromosome 3 were significantly associated with responders. A phase II neoadjuvant PD-1 inhibitor study of 20 cSCC patients reported significantly higher RNA expression of IFNg-related immune genes in the tumors of pathologic responders [24]. Studies have correlated DNA damage repair (DDR) gene mutations to higher TMB levels across five different tumor types; however, there are limited reports evaluating their association to response to checkpoint inhibitors [25]. Specifically for mismatch repair (MMR) genes, a retrospective study of 26 cSCC patients treated with a PD-1 inhibitor found *MLH1* and *MSH6* mutations in two responders, and a case report found a *MLH1* mutation in a complete responder [22,26].

Identifying predictors of response can lead to optimization of patient selection and ultimately improve response rates. In this retrospective study, molecular and mutational biomarkers such as PD-L1 and TMB and the mutational profile in tumor and blood were analyzed in patients with cSCC undergoing treatment with PD-1 inhibitors in order to compare with treatment response, describe correlations, and possibly identify potential predictors of response.

## 2. Materials and Methods

### 2.1. Study Cohort

cSCC patients treated with PD-1 inhibitors at Atrium Health Wake Forest Baptist Cancer Center from 1 January 2017 to 7 January 2024 were identified by an electronic medical record search. Patients who were not evaluable for response and those who did not have genomic testing of the tumor or blood were excluded. General demographics, smoking history, Eastern Cooperative Oncology group (ECOG) score, immunosuppression status, location of disease, treatment history, and treatment outcomes were recorded. All clinical data were reported from the time of initiating immunotherapy to most recent follow-up.

### 2.2. Clinical Management and Treatment Response

All patients had histology-verified cSCC. Patients were treated with pembrolizumab 200 mg every 3 weeks, cemiplimab 350 mg every 3 weeks, or nivolumab 200 mg every 2 weeks. Follow-up imaging was generally obtained every five treatments or at 3-month intervals to assess for treatment response. Treatment response was determined by Immune-related Response Evaluation Criteria in Solid Tumors (iRECIST). Using iRECIST guidelines, treatment was continued until confirmed progression at follow-up imaging 4–8 weeks from the date of unconfirmed progression. Patients were treated for a target of one or two years (per patient’s choice), as tolerated, or until progression. Patients were classified as responders if they achieved complete response, partial response for more than 6 months, or stable disease for more than 1 year. PFS was calculated from the date of treatment initiation to the date of progression by iRECIST criteria. OS was calculated from the date of treatment initiation to the date of death or to the date of the last encounter in any medical records. Duration of follow-up was calculated from the date of treatment initiation to the date of a last encounter with a physician at our institution.

### 2.3. Tumor Molecular Profiling

PD-L1 immunohistochemistry testing and comprehensive genomic profiling were performed by FoundationOne testing (Foundation Medicine, Cambridge, MA, USA) and Guardant360 testing (Guardant Health, Palo Alto, CA, USA). PD-L1 was analyzed by the FDA-approved immunohistochemistry 22C3 pharmDx kit, performed commercially by FoundationOne. PD-L1 expression was reported as tumor proportion score (TPS) or combined positive score (CPS). Next-generation sequencing (NGS) was utilized to obtain comprehensive genomic profiling. Formalin-fixed paraffin embedded tumor tissue specimens were sent to FoundationOne for tumor DNA (tDNA) analysis, and peripheral blood samples were sent to Guardant Health for circulating tumor DNA (ctDNA) analysis. Genomic biomarkers of interest included tumor mutation burden (TMB), microsatellite instability status (MSI), and somatic gene alterations of both clinically known and unknown significance. Synonymous alterations were excluded from this analysis. FoundationOne NGS assesses 324 genes, and Guardant360 NGS assesses 74 genes. Tumor tissue samples were collected prior to starting treatment with PD-1 inhibitors. Circulating tumor DNA blood samples were collected prior to starting treatment with PD-1 inhibitors and throughout the treatment course. The individual gene mutations as well as the group of DNA damage repair (DDR) genes were analyzed in the total patient population and separately in responders and non-responders. The DDR groups included base excision repair (*MUTYH*, *PARP1*), damage sensor (*ATM*, *ATR*, *CHEK1*, *CHEK2*), Fanconi anemia (*FANCA*, *FANCC*, *FANCG*, *FANCL*, *XRCC2*), homologous recombination (*BARD1*, *BRCA1*, *BRCA2*, *BRIP1*, *NBN*, *PALB2*, *RAD51B*, *RAD51C*, *RAD51D*, *RAD52*, *RAD54L*), mismatch repair (*MLH1*, *MSH2*, *MSH3*, *MSH6*, *PMS2*), nucleotide excision repair (*ERCC4*, *POLE*), SWI/SNF chromatin remodeling (*ARID1A*, *SMARCA4*), and TP53 pathway (*MDM2*, *MDM4*, *TP53*), as described previously in the literature [25].

### 2.4. Statistical Analysis

Statistical analysis included the following numeric variables: “PFS (months)”, “OS (months)”, “PD-L1 (Combined)”, “PD-L1 (TPS)”, “PD-L1 (CPS)”, “TMB (tumor)”, and “TMB (blood)”, along with binary variables of 325 genes (combined blood and tissue). For each numeric variable, the data were split into two groups corresponding to responders and non-responders. All statistical computations, machine learning models, and visualizations were generated using Python 3.11.4 and R version 4.4.0 (24 April 2024). The Shapiro–Wilk test was performed for normality in each group, and Levene’s test was performed for homogeneity of variances. Based on the results of these tests, either Welch’s *t*-test or the Mann–Whitney U test were applied. For the binary gene variables, Fisher’s exact test was applied. In the multivariate analysis, Lasso logistic regression was first used to rank the predictive variables in order of feature importance. Traditional logistic regression was initially attempted using these identified variables; however, due to sample size limitations, Firth’s penalized logistic regression was applied. For this dataset size (n = 25), the optimal number of predictors in the logistic regression model is expected to be 2 or 3. A support vector classifier (SVC) was built using eight important genes as features and responder status as the target variable. Data were split into a training set (70%) and a test set (30%). The SVC model was built with the parameters {‘C’: 10, ‘gamma’: 0.1, ‘kernel’: ‘rbf’} and leave-one-out cross-validation (LOO-CV) with the parameters {‘C’: 1, ‘kernel’: ‘linear’}. Principal component analysis (PCA) was used to evaluate performance.

## 3. Results

### 3.1. Demographics and Tumor Characteristics

Between 1 January 2017 and 7 January 2024, 25 cSCC patients treated with a PD-1 inhibitor were identified as evaluable for response and had available genomic data. All patients had locally advanced, unresectable, recurrent, and/or metastatic cSCC. Patients’ age, gender, ECOG performance status, smoking history, immunosuppressive status, tumor location, and previous treatments are presented in Table 1. None of the analyzed patient and tumor characteristics significantly correlated with response to therapy with PD-1 inhibitors (Table 1).

### 3.2. Treatment Results

All 25 patients received monotherapy with either cemiplimab (11 patients), pembrolizumab (12 patients), or nivolumab (2 patients). The average number of treatment administrations was 19 (range: 4–50). Five patients (20%) stopped treatment due to adverse effects: pancreatitis, vertigo, colitis (one patient each), and dermatitis (two patients). Five patients (20%) stopped treatment due to progression of disease. The objective response rate (ORR) was 80% (20/25 patients); ten (40%) patients achieved complete response (CR), nine (36%) achieved partial response (PR) with a duration longer than six months, and one (4%) achieved stable disease (SD) for more than one year. Six out of nine partial responders had near-complete response: four of these six patients stopped treatment prematurely (patient’s disease, unrelated death in two patients, and diagnosis of a second primary cancer that required a different treatment in one patient), and two patients are continuing treatment and have clinical CR, not yet confirmed by imaging. Five (20%) patients did not respond and progressed on treatment. The one patient with a tumor localized outside of the head and neck region (in a limb) was a non-responder.

Seven patients (the five non-responders, one with CR and one with PR) underwent next-line treatments: surgery, surgery and radiotherapy, cetuximab and radiotherapy, or chemoradiotherapy. The one complete responder among these seven patients had recurrence at 35 months post-treatment and is currently under treatment with the same PD-1 inhibitor, with partial response (the second round of PD-1 inhibitor was excluded from this analysis). The one partial responder among these seven patients stopped treatment due to immune-related dermatitis and progressed off treatment at 14 months.

Four out of ten complete responders have more than two years of follow-up post-treatment and remain in remission. Thirteen out of twenty-five (52%) patients are still alive and continue to follow-up. Four patients were lost to follow-up, and at last visit, two were in CR and two were in PR.

The median follow-up for all patients was 21 months (range: 6–61 months). The median PFS was 18 months for all patients (range: 2–61 months)—28 months in responders (range: 6–61 months) vs. 3 months in non-responders (range: 2–8 months) (*p* = 0.00001). The median OS was 21 months for all patients (range: 6–61 months)—30 months in responders (range: 6–61 months) vs. 17 months in non-responders (range: 8–53 months) (*p* = 0.34) (Figure 1).

### 3.3. Molecular and Genomic Results

PD-L1 expression was tested in 21 out of 25 patients and was evaluated either by TPS (16 patients) or CPS scoring (5 patients). Median PD-L1 TPS scoring was 20% (range: 0–100%), and median PD-L1 CPS scoring was 30% (range: 5–100%). PD-L1 score tested by either TPS or CPS was 25% in responders (16 patients, range: 0–90%) compared to 10% (5 patients, range: 1–100%) in non-responders (*p* = 0.39). PD-L1 analyzed by TPS only showed a median PD-L1 of 20% in responders (13 patients, range: 0–100%) and 1% in non-responders (3 patients, range: 1–10%) (*p* = 0.03). PD-L1 analyzed by CPS only showed a median PD-L1 of 30% in responders (three patients, range: 1–90%) and 65% in non-responders (two patients, range: 30–100) (*p* = 0.39) (Figure 2).

TMB levels were tested in tumor and/or in blood. All 25 patients had at least one form of testing completed. Median tumor TMB was 68 mut/Mb (20 patients, range: 4–288 mut/Mb), and median blood TMB was 8 mut/mb (13 patients, range: 2–24 mut/Mb). Analysis of responders compared to non-responders showed a median tumor TMB of 68 mut/Mb (15 patients, range: 4–288 mut/Mb) and 70 mut/Mb (5 patients, range: 8–200 mut/Mb) (*p* = 0.59), respectively, and a median circulating TMB of 10 mut/Mb (11 patients, range: 3–24 mut/Mb) and 5 mut/Mb (2 patients, range: 4–6 mut/Mb), respectively (*p* = 0.08).

MSI status testing was completed for all 25 patients. Two patients were classified as MSI high, one a complete responder and the other one classified as a non-responder with only 8 months of stable disease.

NGS was available for all 25 patients in tumor (20 patients) or blood (23 patients), with 18 patients having both test results available. Out of 324 analyzed genes, there were 277 unique genes that were mutated in this cohort. The most frequently mutated genes in this cohort, present either in tumor, blood, or both, were *TP53* (in 92% patients), *NOTCH1* (in 84% patients), *TERT* (in 72% patients), *CDKN2A* (in 60% patients), *MLL2* (in 56% patients) NOTCH2 (in 48% patients), BRCA2 (in 44% patients), and HNF1A (in 44% patients) (Figure 3).

Eight genes were significantly more frequently mutated in non-responders: *CDK12*, *CTCF*, *CTNNB1*, *IGF1R*, *IKBKE*, *MLH1*, *QKI*, and *TIPARP* (Table 2). Of these eight genes, *CDK12*, *CTNNB1*, and *MLH1* were tested in both tDNA and ctDNA, whereas *CTCF*, *IGF1R*, *IKBKE*, *QKI*, and *TIPARP* were only tested in tDNA. In non-responders, only *CDK12* was found to be mutated in both tDNA and ctDNA, whereas CTNNB1 and MLH1 were only mutated in tDNA. Thirty-one out of thirty-three alterations were point mutations resulting in amino acid changes, frameshift alterations, splice site alteration, or stop codons; one was a rearrangement; and one was an amplification (Appendix A). In a separate analysis of DDR genes, mutations in the MMR gene group were statistically associated with non-responders (*p* = 0.04), whereas mutations in other gene groups classified as DDR genes were not significantly correlated with response (Table 2).

### 3.4. Multivariate and Support Vector Classifier Results

To assess the predictive value of the eight genes that were significantly more prevalent in non-responders, multivariate analysis was performed, and support vector classifier models were built. Traditional logistic regression was initially applied for multivariate analysis but failed in the context of a small sample size due to perfect prediction of the outcome variable by one or more predictors. To address this issue, Lasso logistic regression was applied to identify the feature importance of these eight genes and revealed that *CDK12*, *IGF1R*, *MLH1*, and *TIPARP* most strongly correlated with response. These four genes were further analyzed in two- and three-gene combinations using Firth’s penalized logistic regression. Each gene was significantly associated with decreased odds of response in at least one combination (Table 3).

A support vector classifier model was built using the eight genes as features and responder status as the target variable. Using leave-one-out cross-validation (LOO-CV) and principal component analysis (PCA) to account for small sample size, the model obtained an accuracy of 0.88 in the training set and of 1.0 in the testing set. Shapley values quantify feature importance in machine learning models and prioritized the eight genes of interest in the following order: *MLH1* (+0.11), *TIPARP* (+0.07), *CTCF* (+0.07), *CDK12* (+0.02), *CTNNB1* (+0.02), *IGF1R* (+0.02), *IKBKE* (+0.02), and *QKI* (0) (Figure 4).

## 4. Discussion

The emergence of PD-1 inhibitors has dramatically changed oncology and has provided a new treatment avenue for many patients with advanced malignancies. Treatment outcomes have been remarkable for duration of response, yet one of the continued issues remains limited rate of response. One approach to addressing this limitation is by identifying biomarkers predictive of treatment response, which would allow for better patient selection.

This study identified notable differences between responders and non-responders in a retrospective series of 25 patients with advanced, unresectable, recurrent, and/or metastatic cSCC who were treated with PD-1 inhibitor monotherapy. Compared with other studies, this cohort presents a remarkably higher objective response rate of 80%, almost double compared to prior phase I, phase II, and retrospective studies in cSCC [8,9,21,22]. 40% of patients mounted a CR, which is strikingly greater than the 13% reported in the initial clinical trials. Of the total, 24% of patients achieved major PR with near-CR and may have achieved CR if they had continued treatment. Of the patients in this cohort, 24 out of 25 (96%) had primary disease located in the head and neck region. Multiple studies have reported that cSCC of the head and neck is associated with better outcomes, and this cohort further confirms the correlation [15,17].

Different from previous studies, this analysis considered SD for more than 1 year among responders, whereas other investigators have defined responders as patients with CR or PR. However, this difference in responder definition is unlikely to account for the higher response rate observed in our as study, as just one responder exhibited prolonged SD rather than CR or PR. As opposed to previous studies of PD-1 inhibitors in cSCC, our study used iRECIST rather than RECIST 1.1 to assess radiologic response. Notably, the phase II EMPOWER-cSCC trial described two patients with pseudo-progression and subsequent treatment response who were indeed misclassified as non-responders per the study protocol [9]. Three responders in this study similarly experienced progression by RECIST 1.1 but subsequently had objective response that our use of the iRECIST criteria allowed for us to correctly identify.

Importantly, four out of ten patients who achieved CR remained disease-free for more than two years after completing treatment, which adds to the body of literature suggesting the possibility of cure with PD-1 inhibitors in cSCC [27]. PFS was expectedly significantly longer in responders vs. non-responders. However, OS was not significantly different, likely because many of the non-responders and some of the patients with PR pursued other lines of treatment that eventually conveyed a survival benefit, though were also more toxic. At the same time, several of the patients in CR died of non-cancer-related co-morbidities, which are common in this elderly patient population, with an average age of 77 years. We identified no clinical parameters in this cohort that were significantly associated with treatment response.

PD-L1 expression has had varying degrees of reported value for predicting treatment response, likely due to inconsistent measurement methods and undefined cutoff levels across different cancers. Expression levels are highly influenced by the tumor microenvironment (TME), testing variability, and tumor heterogeneity, thus making it difficult to implement a standard practice [28,29]. Interpretation has been tailored to each individual type of cancer; however, for cSCC, there is not yet an established consensus. In the initial phase II EMPOWER-cSCC trial, treatment response was independent of PD-L1 expression, as responders were equally found to have PD-L1 (TPS) expression <1% or >50% [19]. However, in the pembrolizumab phase II CARSKIN trial, PD-L1 (TPS) expression ≥1% was associated with positive treatment outcomes [20]. Similar to the CARSKIN study, PD-L1 expression measured in our cohort by TPS (18 patients) among responders was significantly higher compared to non-responders. Analysis by CPS only was not statistically significant, limited to just five patients. Similarly, when combining TPS and CPS data, the statistical significance was negated. Since 24 out of 25 patients were PD-L1 positive, no validation could be made regarding predictive value of PD-L1 ≥1%. More investigations are needed to discern predictive thresholds, but preliminary results are available to support the use of a TPS scoring system in cSCC.

TMB is an emerging biomarker and has been interpreted as a surrogate measurement of a tumor’s mutational frequency and likelihood for expressing neoantigens [30,31]. It is postulated that higher TMB is associated with increased immunogenicity, providing a better target for the immune system, which has been shown to correlate with better objective response rates in patients treated with PD-1/PD-L1 inhibitors [32,33]. cSCC is the solid tumor known to have the highest mutational rate, which in part led to the rationale for PD1-inhibitor use. Although TMB is not yet validated in cSCC, Hanna et al. reported a median TMB of 25 mut/Mb in responders compared to 11 mut/Mb in non-responders (*p* = 0.02) [21]. Similarly, In et al. reported TMB of 60 mut/Mb compared to 9 mut/Mb in non-responders (0.04) [22]. Interestingly, in this cohort, non-responders had higher median tumor TMB than responders (*p* = 0.59), with one reaching very high levels up to 200 mut/Mb; however, non-responders had a lower median level of blood TMB (13 patients), though was not statistically significant (*p* = 0.08). Noticeably, the median tumor TMB level was remarkably higher than in blood TMB (68 vs. 8). This discordance suggests a possible difference in the utility of tumor and blood TMB for predicting response outcomes, and further studies are needed to evaluate this possibility.

Tumors with MSI are reported to carry a higher frequency of mutations and thus better responses to PD-1 inhibitors [34,35]. These findings led to FDA approval of pembrolizumab for patients with high MSI, independent of tumor type. In this cohort, two out of twenty-five patients expressed high MSI, one of whom achieved CR and the other one achieved SD for 8 months (classified as non-responder). When analyzing DNA genomic profiles, seven out of nine patients harbored MMR gene mutations but did not display high MSI. Compared to responders, non-responders were associated with increased *MLH1* gene mutations (*p* = 0.03) and with increased mutations in the MMR genes group (*p* = 0.04). These findings differ from reports from In et al., who analyzed 15 cSCC patients treated with PD-1 inhibitor and reported 1 responder with an *MLH1* mutation and another responder with an *MSH6* mutation. They did not report any mutations in non-responders and did not report patients’ MSI status. Prior to PD-1 inhibitor approval in cSCC, Assam et al. reported an unresectable cSCC tumor with *MLH1* loss-of-function that achieved CR with pembrolizumab. These discrepancies in reports are likely attributable to the low sample sizes but also may be a reflection of discordance between genotype and phenotype. MSI and deficient MMR are of continued interest, and larger sample sizes are needed to validate their predictive values.

The use of various known biomarkers has been inconsistent across other studies, and none have been validated as a predictor. Genomic profiles and signatures are becoming a topic of interest for identifying predictors of response. There are limited published data reporting genomic alterations associated with response to PD-1 inhibitor treatment in cSCC patients. In a phase II neoadjuvant cemiplimab trial, responders were found to have significantly higher RNA expression of the IFNg and immune related genes *IDO1*, *CXCL9*, *CD274*, *GZMK*, and *ICOS*, indicating an inflamed TME that may respond more favorably to PD-1 inhibitors [24]. Additionally, in a retrospective study of 33 cSCC patients treated with a PD-1 inhibitor, responders were found to have low-copy gains in the chromosome 3q21-27 region containing *ETV5*, *PIK3CA*, and *BCL6* [23]. In the current study, mutations in eight genes were significantly more prevalent in non-responders compared to responders. The mutations in these genes of interest were mostly identified in tDNA: *CDK12*, *CTCF*, *CTNNB1*, *IGF1R*, *IKBKE*, *MLH1*, *QKI*, and *TIPARP*. Of these genes, only *CDK12*, *CTNNB1*, and *MLH1* were tested for in both tDNA and ctDNA. One non-responder had *CDK12* mutated in both tDNA and ctDNA, whereas *CTNNB1* and *MLH1* mutations were only found in tDNA. All of these eight genes encode proteins with functions in tumor immune response and in the TME. *CDK12* is believed to play a role in the TME by promoting immunosuppressive chemokines and evasion of immune surveillance. This theoretical foundation has inspired the combination of CDK12 inhibitors and immunotherapies [36]. *CTNNB1* encodes for a beta-catenin involved in cell–cell adhesion. In the TME, it is believed to negatively impact immune infiltration and is postulated to be a negative biomarker for checkpoint inhibitor response in hepatic cellular carcinoma [37,38]. CTCF, a chromatin regulator, interacts with TCF-1, a transcription factor, to promote CD8^+^ T cell development and maturation [39,40]. Findings from previous studies in lung cancer suggest that *IGF1R* confers immunotherapy resistance by sustaining tumor immunosuppression [41,42]. *TBK1* has been identified as an immune evasion gene, and it is suspected that its targeting enhances the response to PD-1 inhibition [43]. Both IKBKE, a kinase in the NFkb pathway, and TIPARP, a poly ADP ribose polymerase (PARP), interact with TBK1, and disturbances in their interactions can alter cell mediated IFNg antitumor responses [44,45]. *QKI* encodes for RNA-binding proteins, and there are minimal data about its impact in the TME. *MLH1* is an MMR gene but also may be linked to suppressed immune cell infiltration [46]. These eight genes are found to be mutated in a variety of cancer types and are further described in various pan-cancer studies (Appendix A).

Predictive models were attempted utilizing the eight mutated genes significantly associated with non-responders. Firth’s penalized logistic regression and SVC models were used to mitigate the biases and imbalances associated with small sample size and rare events. Using Lasso logistic regression to rank the importance of each mutated gene, *CDK12*, *IGF1R*, *MLH1*, and *TIPARP* were isolated as the four mutated genes that correlated the strongest with absence of response to treatment. Firth’s penalized logistic regression then was used to evaluate *CDK12*, *IGF1R*, *MLH1*, and *TIPARP* in two-gene and three-gene combinations, and all were observed to be statistically significant in at least one combination. Of note, *CDK12* and *TIPARP* maintained significance most frequently when combined with other gene mutations. These findings from the logistic regression model validate the strength of *CDK12*, *IGF1R*, *MLH1*, and *TIPARP* as potential predictors for treatment response, with a possible emphasis on *CDK12* and *TIPARP*. Given the small sample size of this cohort, the eight genes of interest were separately evaluated in a machine learning SVC model to verify the findings. When testing their combined predictive value, the eight genes of interest performed exceptionally well together. By applying importance on these 8 mutated genes out of 277 mutated genes, the SVC model classified responders and non-responders with an accuracy of 88% in the training dataset. Learning from this training dataset, the system was able to precisely classify responders and non-responders with an accuracy of 100% in the testing dataset. This observed performance validates the association of mutations in these eight genes with absence of response to treatment and supports their capability of classifying patients into responders vs. non-responders based on their genomic profile. When assessing the contributing strength of each gene mutation, the model prioritized them in the following order: *MLH1*, *TIPARP*, *CTCF*, *CDK12*, *CTNNB1*, *IGF1R*, *IKBKE*, and *QKI* (provided zero contribution in the model). Notably, three of the four strongest contributors (*MLH1*, *TIPARP*, and *CDK12*) also correlated with non-response in the logistic regression model. *QKI* did not have any impact in either model. Additionally, both models generated high cross-validation scores, which indicates generalizability to unseen data. The concordance between these two models verifies the predictive value of these mutated genes and highlights *CDK12*, *MLH1*, and *TIPARP* as the genes most appropriate for further study. Overall, the majority of these mutated genes have a commonality of CD8^+^ T cells and IFNg effects at the TME level.

This analysis is limited by its small and unbalanced sample size, which increases the likelihood of observations of chance and the potential for model overfitting to the dataset. However, the functional commonality of these genes increases the likelihood of reproducible and generalizable findings. This study did not investigate the phenotypic consequences of genomic alterations, including downstream effects on RNA expression, proteomics, gain of function, or loss of function. These additional types of testing may further elucidate the mechanisms of the predictive genomic biomarkers identified by this study. Additionally, the tDNA sequencing and ctDNA sequencing used in this study did not test for all of the same genes, which may explain why most were found in tDNA. Among the eight significant mutated genes, *CDK12*, *CTNNB1*, and *MLH1* were tested in both tumor and blood, whereas *CTCF*, *IGF1R*, *IKBKE*, *QKI*, and *TIPARP* were only tested in tDNA. Nevertheless, this study identified eight genes that are significantly correlated with PD-1 inhibitor non-responder status in cSCC and that may be useful in the future as predictive biomarkers.

## 5. Conclusions

This study demonstrated that PD-1 inhibitors can achieve remarkable response rates and durable PFS in cSCC, especially when tumors are located in the head and neck region. This study further described the mutational profile in tumor tissue and blood for this type of cancer, and the investigation of genomic biomarkers for prediction of response to PD-1 inhibitors revealed a group of eight mutated genes that significantly correlated with non-responder status. Predictive models by logistic regression and SVC were concordant, verifying the predictive value of these gene mutations. Biologic functions of the identified genes involve immune cell inflammatory responses and their interactions with the TME. Analysis of PD-L1 as a biomarker of response to PD-1 inhibitors monotherapy showed significant predictive ability only when measured by TPS, while TMB did not predict treatment response. These novel genomic biomarkers highlight biologic pathways that may be of increased interest to other researchers, and their ability to predict response to PD-1 inhibitors needs to be validated in further large studies.

## Figures and Tables

**Figure 1 cancers-17-01172-f001:**
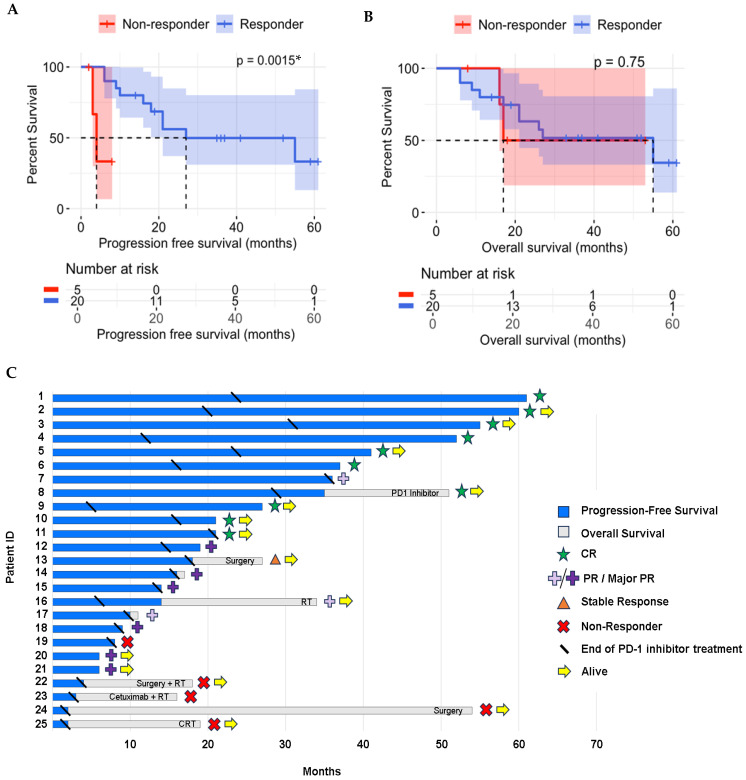
(**A**) Progression-free survival for all patients. (**B**) Overall survival for all patients. (**C**) Swimmer’s plot describing clinical milestones of the 25 cSCC patients treated with PD-1 inhibitor: progression-free survival, overall survival, complete response, partial response/major partial response, stable response, non-responder, end of PD-1 inhibitor treatment, alive. CR: complete response; CRT: chemotherapy and radiation therapy; PR: partial response; RT: radiation therapy; (*) = statistically significant at *p* < 0.05 using Welch’s *t*-test.

**Figure 2 cancers-17-01172-f002:**
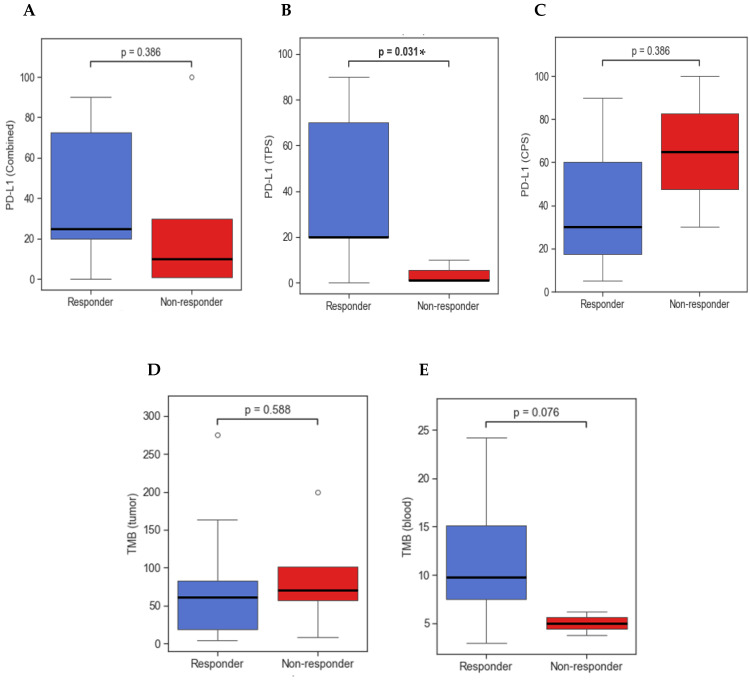
(**A**) PD-L1 expression (by TPS or CPS) in responders compared to non-responders. (**B**) PD-L1 expression (by TPS) in responders compared to non-responders. (**C**) PD-L1 expression (by CPS) in responders compared to non-responders. (**D**) Tumor tissue TMB levels in responders compared to non-responders. (**E**) Circulating tumor TMB levels in responders compared to non-responders. TMB: tumor mutational burden; TPS: tumor proportion score; CPS: combined positive score. (*) = statistically significant at *p* < 0.05 using Welch’s *t*-test or Mann–Whitney U Test.

**Figure 3 cancers-17-01172-f003:**
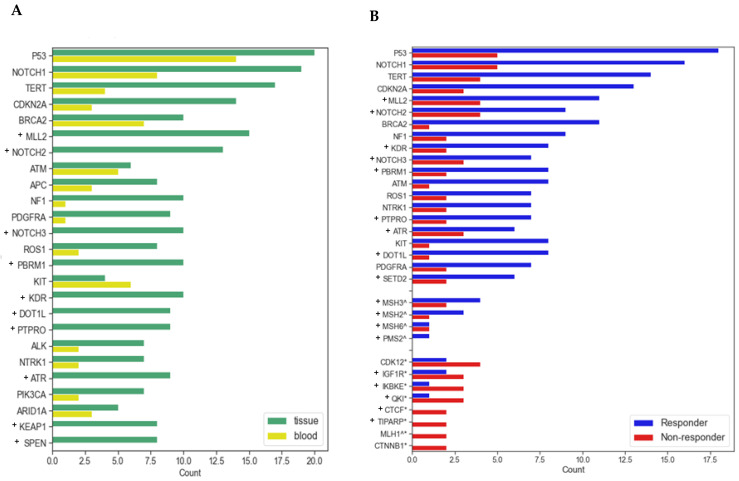
(**A**) Incidence of most frequently mutated genes in tumor tissue and blood in all patients. (**B**) Incidence of most frequently mutated genes in responders and non-responders. (^) = mismatch repair genes. (*) = genes significantly more mutated in non-responders. (+) = tested by FoundationOne in tissue but not Guardant360 in blood.

**Figure 4 cancers-17-01172-f004:**
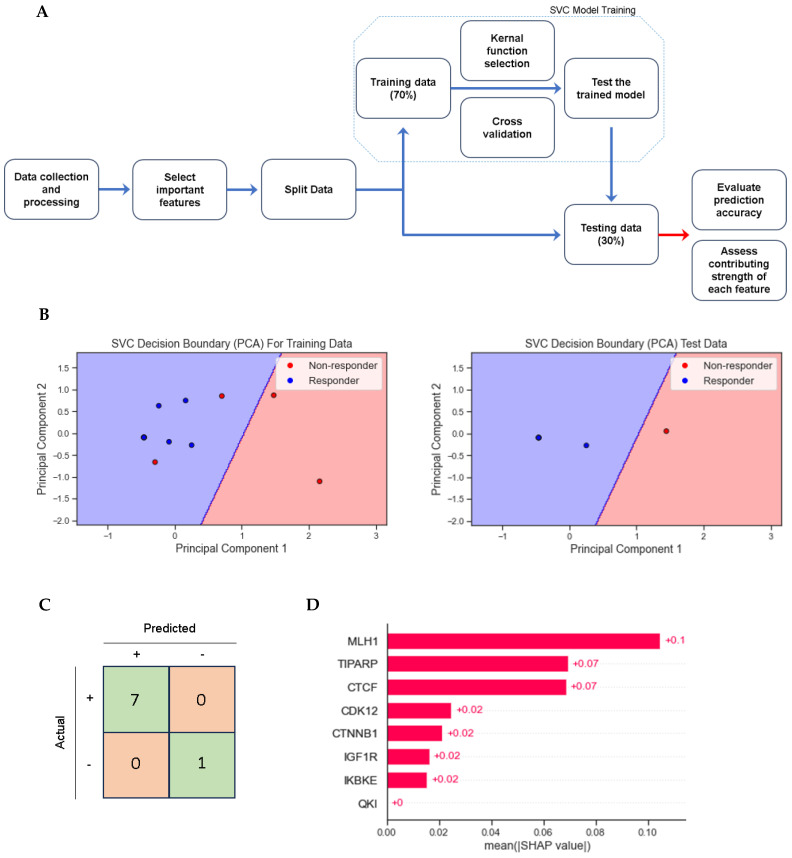
(**A**) Schematic representation of the process of building an SVC model. (**B**) Results of SVC training dataset classification and test dataset classification. Blue area predicts response, red area predicts no response, blue dot depicts “observed responder”, red dot depicts “observed non-responder”. (**C**) Confusion matrix depicting accuracy of SVC performance on test dataset classification. (**D**) Shapley values indicating the predictive strength and contribution of each individual gene within the SVC model. SVC: support vector classifier; PCA: principal component analysis.

**Table 1 cancers-17-01172-t001:** Demographics and clinical characteristics of cSCC patients at the start of treatment with PD-1 inhibitor.

	All Patients		Responders		Non-Responders		(*p*-Value *)
	(n = 25)		(n = 20)		(n = 5)		
**Age (range)**	77	(58–97)	78	(59–97)	75	(58–92)	0.73
**Gender**							0.53
Male	20	80%	17	85%	3	60%	
Female	5	20%	3	15%	2	40%	
**ECOG performance status**							0.29
0	2	8%	1	5%	1	20%	
1	15	60%	12	60%	3	60%	
2	6	24%	6	30%	0	0%	
3	2	8%	1	5%	1	20%	
**Smoking history**							0.48
Never smoker	12	48%	10	50%	2	40%	
Former smoker	10	40%	7	35%	3	60%	
Current smoker	3	12%	3	15%	0	0%	
**Immunosuppression**							0.65
None	22	88%	17	85%	17	85%	
Heme malignancy	2	8%	2	10%	2	10%	
Autoimmune disorder	1	4%	1	5%	1	0%	
**Primary site of disease**							0.33
Face	5	20%	5	25%	0	0%	
Scalp	9	36%	7	35%	2	40%	
Eye orbit	1	4%	1	5%	0	0%	
Ear	3	12%	2	10%	1	20%	
Nose	2	8%	2	10%	0	0%	
Neck	3	12%	3	15%	0	0%	
Extremity	1	4%	0	0%	1	20%	
**Cancer status**							0.39
LRA stage III or IV	21	84%	16	80%	5	100%	
Recurrent	16	64%	14	70%	2	40%	
Distant metastasis	4	16%	4	20%	2	0%	
**Prior treatment**							0.66
None	5	20%	5	25%	0	0%	
Surgery alone	11	44%	8	40%	3	60%	
RT alone	1	4%	1	5%	0	0%	
Chemo alone	0	0%	0	0%	0	0%	
Surgery + RT	7	28%	5	25%	2	40%	
RT + chemo	0	0%	0	0%	0	0%	
Surgery + chemo	0	0%	0	0%	0	0%	
Surgery + RT + chemo	1	4%	1	5%	0	0%	
**Treatment with PD-1 inhibitor**							
Average # of treatments	19		22		6		
Cemiplimab	11	44%	8	40%	3	60%	
Nivolumab	2	8%	2	10%	0	0%	
Pembrolizumab	12	48%	10	50%	2	40%	
**Genomic testing availability**							
ctDNA	23	96%	18	90%	5	100%	
tDNA	20	80%	15	65%	5	100%	
Both	18	64%	13	55%	5	100%	

cSCC: cutaneous squamous cell carcinoma; ctDNA: circulating tumor DNA; ECOG: Eastern Cooperative Oncology Group; LRA: locoregionally advanced; RT: radiation therapy; tDNA: tumor tissue DNA; TMB: tumor mutational burden; mut/mb: mutations/megabase. (*) = statistically significant at *p* < 0.05; Wilcoxon rank-sum test was used for continuous variable. Chi squared test was used for categorical variables.

**Table 2 cancers-17-01172-t002:** Gene mutations significantly more prevalent in non-responders (**A**) and impact of DDR gene groups on response rates using multivariate analysis of variance (**B**).

**A**			
**Gene**	**Responders**	**Non-Responders**	**(*p*-Value *)**
*CDK12*	2/20	4/5	(0.005 *)
*CTCF*	0/20	2/5	(0.033 *)
*CTNNB1*	0/20	2/5	(0.033 *)
*IGF1R*	2/20	3/5	(0.038 *)
*IKBKE*	1/20	3/5	(0.016 *)
*MLH1*	0/20	2/5	(0.033 *)
*QKI*	1/20	3/5	(0.016 *)
*TIPARP*	0/20	2/5	(0.033 *)
**B**			
**DDR Gene Groups**	**(*p*-Value *)**	**Mismatch Repair**	
Base excision repair	-		
Damage sensor	(0.5) ^+^		
Fanconi anemia	(0.32)		
Homologous recombination	(0.22)		
Mismatch repair	(0.04 *)	*MLH1*	(0.03 *)
		*MSH2*	(1)
		*MSH3*	(0.56)
		*MSH6*	(0.37)
		*MS2*	(1)
Nucleotide excision	(0.13)		
SWI/SNF chromatin remodeling	(0.32)		
TP53 pathway	(0.15)		

Base excision repair genes group includes *MUYTH*, *PARP1*; damage sensor genes group includes *ATM*, *ATR, CHEK1*, *CHEK2*; Fanconi anemia genes group includes *FANCA, FANCAC*, *FANCG, FANCL, XRCC2*; homologous recombination genes group includes *BARD1*, *BRCA1*, *BRCA2, BRIP1*, *NBN*, *PALB2*, *RAD51B*, *RAD51C*, *RAD51D*, *RAD52*, *RAD54L*; mismatch repair genes group includes *MLH1*, *MSH2*, *MSH3*, *MSH6*, *PMS2*; nucleotide excision repair genes group includes *ERCC4*, *POLE*; SWI/SNF; chromatin remodeling genes group includes *ARID1A*, *SMARCA4*; TP53 pathway genes group includes *MDM2*, *MDM4*, *TP53.* DDR: DNA damage repair; SWI/SNF: SWItch/Sucrose Non-Fermentable. (*) = statistical significance at *p* < 0.05 using multivariate analysis of variance (MANOVA). (^+^) = “*CHEK1*” removed from group analysis as there were no values present. (-) = not assessable since data are insufficient.

**Table 3 cancers-17-01172-t003:** Multivariate analysis of significant gene mutations’ impact on treatment response.

Three-Gene Combinations	OR	95%CI	(*p*-Value *)	CV Score
*CDK12*	0.05	(−8.05, −0.91)	0.003 *	0.97
*TIPARP*	0.003	(−16.03, −1.44)	0.009 *	
*IGF1R*	0.04	(−11.87, −0.23)	0.063	
*CDK12*	0.15	(−4.3, −0.26)	0.023 *	0.97
*MLH1*	0.39	(−6.29, 2.86)	0.547	
*IGF1R*	0.22	(−4.55, 1.42)	0.282	
*TIPARP*	0.03	(−8.59, −0.82)	0.010 *	0.97
*MLH1*	0.03	(−10.14, −0.27)	0.067	
*IGF1R*	0.61	(−3.95, 4.98)	0.766	
*CDK12*	0.09	(−7.19, −0.42)	0.018 *	0.93
*TIPARP*	0.01	(−13.29, −1.37)	0.005 *	
*MLH1*	0.04	(−8.42, 0.14)	0.054	
**Two-Gene Combinations**	**OR**	**95%CI**	**(*p*-Value *)**	**CV Score**
*TIPARP*	0.01	(−9.35, −1.29)	0.004 *	0.97
*MLH1*	0.01	(−9.35, −1.29)	0.004 *	
*CDK12*	0.12	(−4.67, −0.51)	0.009 *	0.97
*IGF1R*	0.13	(−4.87, 0.44)	0.101	
*CDK12*	0.13	(−4.74, −0.31)	0.020 *	0.93
*MLH1*	0.12	(−7.22, 1.04)	0.181	
*CDK12*	0.03	(−8.54, −0.99)	0.002 *	0.82
*TIPARP*	0.01	(−10.49, −0.9)	0.013 *	
*MLH1*	0.12	(−7.33, 1.12)	0.203	0.8
*IGF1R*	0.23	(−3.97, 1.15)	0.244	
*TIPARP*	0.03	(−8.73, −0.32)	0.030 *	0.77
*IGF1R*	0.09	(−5.2, −0.09)	0.042 *	

Treatment response is the outcome variable for each combination of predictive factors. Significant genes were combined into triples and pairs. CI: confidence interval; CV score: cross-validation score; OR: odds ratio. (*) = statistical significance at *p* < 0.05 using Firth’s penalized logistic regression.

## Data Availability

The datasets generated and analyzed are available from the corresponding author upon request.

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
