# Peer review of "Mutational Profile of Blood and Tumor Tissue and Biomarkers of Response to PD-1 Inhibitors in Patients with Cutaneous Squamous Cell Carcinoma"

_cancers, 2025, doi:10.3390/cancers17071172_

Round 1

Reviewer 1 Report

Comments and Suggestions for Authors

Cutaneous squamous cell carcinoma (cSCC) is a prevalent form of skin cancer, representing the second most common type of skin malignancy worldwide. Management of cSCC typically involves surgical excision, but can also require more aggressive treatment approaches, including radiation therapy or immunotherapy wit PD-1 inhibitors.

The identification of biomarkers that predict responses to immunotherapy in cSCC is critical for optimizing treatment strategies and improving patient outcomes. Recent studies have highlighted several potential biomarkers, including tumor mutational burden (TMB), programmed death-ligand 1 (PD-L1) expression, and various inflammatory markers.

The authors performed a retrospective study on tissue and blood samples of 25 cSCC patients undergoing therapy with PD-1 inhibitors. 

The authors performed a complete job both in terms of parameters analyzed and statistical methods used.

Their results highlight how PDL-1 expression and TMB alone fail to be significant in discriminating responder patients from non-responders, while the investigation of genomic biomarkers for prediction of response to PD-1 inhibitors revealed a group of eight mutated genes that significantly correlated with non-responder status.

These new biomarkers useful for the selection of patients responsive to PD-1 inhibitors will certainly need to be confirmed by studies with a larger population,  but they still represents a valid approach to the search for therapeutic improvement in cSCC patients.

In conclusion the results of the work are encouraging, given the therapeutic difficulty of severe forms of sSCC: selecting responder patients will allow a targeted and more effective use of PD-1 inhibitors.

Author Response

Thank you for your considerate review of our manuscript. We greatly appreciate your input.

Reviewer 2 Report

Comments and Suggestions for Authors

Peer review on the paper titled “Mutational Profile of Blood and Tumor Tissue and Biomarkers of Response to PD-1 Inhibitors in Patients with Cutaneous Squamous Cell Carcinoma” Journal – MDPI, Cancers.

Brief summary:

This study investigated the genomic profiles of cutaneous squamous cell carcinoma patients treated with a PD-1 checkpoint inhibitor. The authors found that PDL-1 expression and tumor mutational burden (TMB) were not strongly predictive of treatment response. However, they identified eight genes that were significantly more frequently mutated in non-responders compared to responders.

Overall review:

This study provides valuable insights into a set of genes that are specifically mutated in non-responders. The authors employed rigorous and robust statistical analyses to support their findings, which is commendable. However, this remains the primary finding of the paper.

The authors were thorough in reporting patient demographics and clinical characteristics and did an excellent job detailing individual response rates. Nonetheless, incorporating additional analyses or perspectives could enhance the paper’s appeal to readers and increase its utility for other researchers.

Specific comments:

Issues to address:

  • Do the eight frequently mutated genes in non-responders exhibit any specific patterns when analyzed in relation to demographics and clinical characteristics such as age, sex, and prior treatment? If so, it might be good to add an additional figure addressing this analysis.

  • For the eight genes frequently mutated in non-responders, what types of mutations are observed (e.g., point mutations, hotspot mutations)? Additionally, could you include a figure illustrating the distribution of these mutations across individual patients?

  • Are these mutations commonly found in other cancers, or are they specific to cutaneous squamous cell carcinoma (cSCC)? Could you please address this in the text?

Reviewer 3 Report

Comments and Suggestions for Authors

This study ia an interesting and  clinically valuable because  there is a need predictive biomarkers for target therapy.  However,  I  do have some comments :

Introduction, lines 104 – 108 describes the group of patients. It  should be included in  Materials and Methods ( Study cohort).

The aim of this paper should be better presented and relate to the main issue of this study. Treatment results lines 256-288 should be reduced because significant proportion of  results are presented in Table 1

 Line  388, the Authors state that “NGS was available for all 25 patients in tumor (20 patients) or blood (23 patients) and  Figure 3 A showed mutation in tumor tissue and blood in all patients. Line  390-393, the Authors indicate  that  “The most frequently mutated genes in 390 all patients were TP53 ( (23 patients, 92%), NOTCH1 (21 patients, 84%), TERT (18 patients, 391 72%), CDKN2A (15 patients, 60%), MLL2 (14 patients, 56%) NOTCH2 (12 patients, 48%), 392 BRCA2 (11 patients, 44%), and HNF1A (11 patients, 44%).  It is not clear what the results present.  Percentage of patients with mutated gene in tissue or in blood?   Please explain.

Author Response

This study is an interesting and clinically valuable because there is a need predictive biomarkers for target therapy.  However, I do have some comments:

Comment 1: Introduction, lines 104 – 108 describes the group of patients. It should be included in Materials and Methods (Study cohort).

Response 1: Thank you for noticing this shortcoming. We removed the patients’ description from this Introduction paragraph. The inclusion criteria for the study population are described in the Material and Methods and the description of the cohort of patients identified for our study is included in the first paragraph of the Results.

Comment 2: The aim of this paper should be better presented and relate to the main issue of this study. Treatment results lines 256-288 should be reduced because significant proportion of results are presented in Table 1

Response 2: We appreciate the recommendations and changed the last paragraph of the Introduction section to address comment 1 and 2. We also simplified significantly the paragraph describing Table 1, as recommended in Comment 2.

Comment 3: Line  388, the Authors state that “NGS was available for all 25 patients in tumor (20 patients) or blood (23 patients) and  Figure 3 A showed mutation in tumor tissue and blood in all patients. Line  390-393, the Authors indicate  that  “The most frequently mutated genes in all patients were TP53 ( (23 patients, 92%), NOTCH1 (21 patients, 84%), TERT (18 patients, 72%), CDKN2A (15 patients, 60%), MLL2 (14 patients, 56%) NOTCH2 (12 patients, 48%), BRCA2 (11 patients, 44%), and HNF1A (11 patients, 44%).  It is not clear what the results present.  Percentage of patients with mutated gene in tissue or in blood?   Please explain.

Response 3: Thank you for pointing out this confusion. In figure 3A we have changed the caption to specify how many samples were from tumor vs. blood.

In regards to lines 390-393, the percentages represent the percentages of patients who have the respective gene mutated in either blood or tumor or both. These updates may be found in lines 388-390.

Reviewer 4 Report

Comments and Suggestions for Authors

In this study, the authors investigated the response to PD-L1 inhibitors in 25 patients with advanced cSCC. They also compared the molecular profiles and PD-L1 expression between responders and non-responders. The manuscript is well-written and presents a well-organized data analysis. I have a few minor comments.

  1. What clinical stages (TNM) were assigned to the cSCC before treatment? Is there a correlation between staging and the response to PD-L1 inhibitors?
  2. Is there a current guideline regarding the role of PD-L1 immunostaining in patients with cSCC? For instance, is PD-L1 immunostaining required to initiate treatment?
  3. What role does surgery play for patients with partial responses? If tumors become surgically resectable after immunotherapy, is surgery considered the standard secondary treatment?

Minor comments:

  1. Please include the full description of abbreviations used in the abstract.
  2. Line 88, page 2: "In et al. analyzed 15 cSCC patients" — the author’s name should be added before "et al."

Author Response

In this study, the authors investigated the response to PD-L1 inhibitors in 25 patients with advanced cSCC. They also compared the molecular profiles and PD-L1 expression between responders and non-responders. The manuscript is well-written and presents a well-organized data analysis. I have a few minor comments.

Comment 1: What clinical stages (TNM) were assigned to the cSCC before treatment? Is there a correlation between staging and the response to PD-L1 inhibitors?

Response 1: All patients had loco-regionally advanced stages III and IV (added to Table 1) or recurrent/metastatic disease (Please see Table 1 for percentages). There was no statistical correlation between any aspect of staging or tumor status and response to PD-1 inhibitors. We did not include a more detailed description of TNM staging due to wording limitation and lack of correlation with treatment response.

Comment 2: Is there a current guideline regarding the role of PD-L1 immunostaining in patients with cSCC? For instance, is PD-L1 immunostaining required to initiate treatment?

Response 2: Currently there are no guidelines for testing for PD-L1 in patients with cSCC, regarding the type of antibodies to be used, the type of test recommended, or the use of the test result to guide the treatment with PD-1 inhibitors.

Comment 3: What role does surgery play for patients with partial responses? If tumors become surgically resectable after immunotherapy, is surgery considered the standard secondary treatment?

Response 3: There is currently no standard recommendation for the use of surgery for patients who respond to immunotherapy. There are no clinical studies to measure the risk /benefit ratio and the effect on survival. In our population, although majority of patients continued to follow with their surgeons, only one patient who had stable disease after 18 mo of treatment, and one non-responder elected to undergo surgery.

Comment 4: Please include the full description of abbreviations used in the abstract.

Response 4: Thank you for catching these errors. We have made the according adjustments in the abstract to include full descriptions of any abbreviations used. Specifically, CR, PR, SD, and PFS were addressed.

Comment 5: Line 88, page 2: "In et al. analyzed 15 cSCC patients" — the author’s name should be added before "et al."

Response 5:  Thank you for this suggestion. The author’s name is Gino In therefore, no changes will be made.